# Mid-Term Clinical Outcomes Following Drug-Coated Balloons in Coronary Artery Disease

**DOI:** 10.3390/jcm11071859

**Published:** 2022-03-27

**Authors:** Gal Sella, Gera Gandelman, Nicholay Teodorovich, Ortal Tuvali, Omar Ayyad, Haitham Abu Khadija, Dan Haberman, Lion Poles, Michael Jonas, Igor Volodarsky, Jacob George, Alex Blatt

**Affiliations:** Heart Center, Kaplan Medical Center, Rehovot, Hebrew University, Jerusalem 91905, Israel; gera_g@clalit.org.il (G.G.); mrnikolayta@clalit.org.il (N.T.); ortalhaga@clalit.org.il (O.T.); omeray@clalit.org.il (O.A.); haithamab1@clalit.org.il (H.A.K.); danha1@clalit.org.il (D.H.); lion_p@clalit.org.il (L.P.); michaelyo2@clalit.org.il (M.J.); igorv@clalit.org.il (I.V.); kobige@clalit.org.il (J.G.); alexbl31@clalit.org.il (A.B.)

**Keywords:** coronary intervention, neoatherosclerosis, stent thrombosis, TLR, advantages

## Abstract

Objective: The aim of this study was to evaluate the mid-term efficacy of drug-coated balloons (DCB) in percutaneous coronary intervention (PCI) in two different pathophysiologic scenarios. Background: There are different underlying pathological processes in coronary artery disease. Mid-term safety and efficacy of DCB approach is still limited. Methods: Medical records of all consecutive patients undergoing DCB were evaluated. The primary endpoint was the rate of clinically driven target lesion revascularization (TLR) after 24 months. Results: Between January 2011 and December 2017, 442 patients were included, representing 4.4% of all PCIs in our institution. A total of 460 DCB lesions were treated, of which 328 (71.3%) were de novo and 132 (28.7%) were combined bare metal or drug-eluting stents with in-stent restenosis (ISR). The patients’ mean age was 66.2 ± 11.7 years with a diabetes prevalence of 45.3%. The TLR rate was lower in the de novo group (5.3%) compared to the ISR group (9.4%) (*p* = 0.04). No differences were observed in major adverse cardiovascular events (MACE) between the de novo group (38.9%) and ISR group (42.5%) (*p* = 0.47). No significant differences were detected in the TLR occurrence in the subgroup analysis. Conclusion: Our extended experience demonstrates that the mid-term DCB approach in these two pathophysiologic settings represent a reasonable option, with low TLR rate.

## 1. Introduction

In the daily practice of almost all fields of cardiology, we continually deal coronary artery disease (CAD) and its consequences. Despite our familiarity with its clinical aspects, theories of the formation of coronary atherosclerosis evolved in the past few years [1,2] and had an important practical impact concerning the selection of accurate medical care for the patients [3,4]. Understanding of pathophysiological processes involved in coronary atherosclerosis formation will continue to evolve, primarily due to advances in both laboratory tools and clinical practices [2]. Evolutions in cardiovascular medicine [5] allow us to better define the atherothrombotic nature of vascular diseases, ensuring that strategies for the implementation of management and revascularization are becoming more sophisticated and successful [3,4,6].

The treatment of coronary lesions triggers processes in vessel walls with different characteristics depending on the revascularization strategy; post-traumatic healing following plain balloon angioplasty triggers vessel recoil and neointimal overgrowth; bare metal or drug eluting stent implantation results in neointimal cell proliferation, scar tissue formation, and ultimately, the generation of neoatherosclerosis [7,8,9,10,11].

DCBs (drug-coated balloons) were presented two decades ago as an alternative to plain-balloon inflation or subsequent stent implantation in order to overcome in-stent restenosis. This methodology has been adopted in ESC myocardial revascularization guidelines since 2014 as a class I/level A recommendation for the treatment of in-stent restenosis within BMS or DES [12]. Those balloons are coated with the anti-proliferative agent Paclitexal, which has been shown to prevent neointimal growth and hence may prevent the formation of future neo-atherosclerotic plaque and trigger the recurrence of in-stent restenosis [13]. Recently, there has been growing evidence that DCBs can also be used off-label to treat de novo lesions in small vessels [14].

In this study, our aim is to evaluate mid-term clinical efficacy of DCB angioplasty as a stand-alone treatment in the setting of daily work surrounding de novo and ISR lesions in real-world cath-lab.

## 2. Methods

### 2.1. Patient Population and Inclusion and Exclusion Criteria

The study protocol was reviewed and approved by our institutional ethics committee (Helsinki Committee for Human Rights—Kaplan Medical Center, Rehovot, Israel) before the study began. All data were fully anonymized before access and the ethics committee waived the requirement for informed consent.

All consecutive patients in our institution who underwent DCB were retrospectively included. Eligible patients were those with lesions in the coronary vessel tree who planned to be treated with DCB only. The DCB strategy was at the operator discretion according to the clinical and anatomical characteristics.

All lesion locations and characteristics were included. Lesions pre-specified to be stented (i.e., bifurcations) were excluded. Other exclusion criteria were procedures that included a mixture of DCB and stent implantation in different sites and stage PCI patients with an interval of two months. The presence of dissection type D, E, or F, according to the NHLBI classification system or deterioration of TIMI flow to <II, was an indication of bailout stenting. Bailout-stented lesions after DCB were included.

### 2.2. Procedure Description

Femoral or radial access was chosen by the operator using recommended introducer sheaths of at least 6 French. During intervention, close adherence to the ESC guidelines for myocardial revascularization was strongly endorsed. Special emphasis was paid to adequately prepare the coronary lesion prior to DCB treatment. Predilatation with plain uncoated balloons bearing a balloon-to-vessel ratio of 0.8–1.0 and inflation pressures exceeding nominal pressure was mandatory. Subsequently, second-generation paclitaxel-iopromide DCB angioplasty with a SeQuent^®^ Please (B. braun, Melsungen, Germany) or a Pantera LUX (Biotronik, Berlin, Germany) balloon catheter was performed in the absence of a major flow-limiting dissection. Periprocedural and post-procedure medication protocols were as follows: intravenous administration of heparin (70 IU/kg) or 5000 IU was recommended and supplemented when required, acetylsalicylic acid (ASA) 75–325 mg/day life-long, and a clopidogrel loading dose of at least 300 mg complemented with a regimen of 75 mg/day for four weeks were recommended. In the setting of acute coronary syndrome (ACS), 12 months of dual antiplatelet treatment (DAPT) was recommended. Prasugrel or ticagrelor was chosen instead of clopidogrel based on applicable guidelines. Preloading with clopidogrel, ticagrelor, or prasugrel prior to the procedure was acceptable.

### 2.3. Study Definition Criteria for Events

The primary endpoint was the clinically driven target lesion revascularization rate (TLR) at two years. Secondary endpoints were mortality rate at two years, major adverse cardiac events (MACE), defined as the composite of TLR, cardiac death, cardiovascular hospitalizations, and definite vessel thrombosis also after two years of follow-up. MI was defined according the Universal Definition Guidelines [15]. Definite acute/subacute vessel thrombosis was defined according to the ARC criteria [16]. When the cause of death was unknown or undeterminable, it was assumed to be of cardiac origin.

### 2.4. Statistical Analysis

An established electronic data-capture system was used to document all relevant patient and lesion data. The patients’ electronic and regular records were followed at the end of the study. A χ^2^ test or Fisher’s exact test was used to demonstrate differences between treatment groups when dichotomous variables were compared. The unpaired Student’s t test or the Whitney–Mann nonparametric test was applied for continuous variables. To evaluate normal distribution, either the Kolmogoroff–Smirnoff or the Shapiro–Wilk tests were used. For all tests, the significance level α was 0.05. Based on an estimated 80% inclusion rate for the indication of de novo lesions, an overall TLR rate of 6.0% for a DCB-only treatment group was assumed and compared with a literature value of 12%. The one-group χ^2^ test with a 0.025 one-sided significance level will have 80% power to reject the null hypothesis (not inferior to literature TLR rate) when the sample size is 351. With a 20% attrition rate, a total of 409 patients had to be recruited. The biometric estimates were conducted with nQuery/nTerim version 2.0 (Statistical Solutions Ltd., Cork, Ireland), whereas SPSS version 20.0 (IBM, Munich, Germany) was used for all other statistical analyses.

## 3. Results

### 3.1. Patients Characteristics

Between January 2011 and December 2017, 442 patients who underwent DCB were included (Table 1), representing 4.4% of all PCIs in our institution. In the overall study population, the mean patient age was 66.2 ± 11.7 years. Standing out was the higher prevalence of diabetes in the two groups. Hyperlipidemia and previous history of ischemic heart disease were more frequent in the restenosis group. One-third of total DCB angioplasties were performed in the clinical setting of an ACS-non-ST elevation MI or unstable angina pectoris. In 116 patients (36.4%), DCB angioplasty was performed in the clinical setting of an ACS for de novo lesions (Table 1).

### 3.2. Intervention Data

Table 2 summarizes the procedure findings. A total of 460 lesions with an PCI-DCB-only approach were treated, 328 (71.3%) had PCI for treating de novo lesions, and the rest (132, 28.7%) had combined DES and BMS-ISR angioplasty.

For the overall population, the angioplasty with a DCB-only approach was performed in the left anterior descending artery (LAD), circumflex artery (CX), right coronary artery (RCA), intermediate branch, and left main in 186 (42.3.1%), 150 (34.1%), 76 (17.3%), 21 (4.8%), and 7 (1.6%) patients, respectively. The reference vessel diameter and lesion length for the total population were 2.64 ± 0.59 and 22.62 ± 6.52 mm, respectively. The reference vessel diameter was significantly smaller in the de novo lesions group than the ISR group at 2.51 ± 0.53 mm and 2.97 ± 0.58 mm, respectively, *p* < 0.001. In comparison to the de novo group, ISR lesions were not significantly shorter. To treat 460 lesions, 471 DCBs were employed. DCB dimensions matched the reference vessel diameter and exceeded the mean lesion length. Overall technical success was 99.4%. Bailout stent implantation in the overall population was necessary in 61 (13.7%) lesions and significantly more in the de novo (55 (17.3%)) than in the ISR group (6 (4.7%)) (*p* < 0.001). The use of intravascular imaging was very low, less than 4% in the total cohort, since it is not the standard practice in our institution.

### 3.3. Clinical Outcome

The period of time of available clinical information for all included patients was a mean of 23.7 ± 1.8 months. At the two-year follow-up, there was a strong trend toward fewer TLRs in the de novo group (5.3%) compared to the ISR group (10.2%) (*p* = 0.063) (Table 3 and Figure 1). The Kaplan–Meyer survival curves according to the TLR success were overlapping in the de novo group, with an early separation and persistent maintenance over time (Figure 2). The overall MACE rate was 39.9%, with no statistical difference between the two groups (*p* = 0.478). The vessel thrombosis rate was null.

### 3.4. Variables Analysis

We saw no clinical finding that dependently predicts the TLR rate at 24 months in the subgroup analysis (Figure 3).

## 4. Discussion

Our main findings show that PCI-DCB only approach is associated with a low incidence of TLR, namely low clinical-driven TLR, after two-year follow up in the two different pathophysiological settings. The low TLR rate obtained in this study was with the range found in previous similar registries [17,18]. In current daily practice, a stented lesion, principally by DES, is the more prevalent strategy, despite the potential for target lesion failure [19,20] (TLF) and disadvantages such as permanence of durable metallic mesh and/or polymer layer, the fact that the healing process starting with tissue hyperplasia will finish later in neoatherosclerosis, and the loss of vasomotor response properties of the vessels. In this respect, DCB and biodegradable stents have clear advantages [21,22,23].

The purpose of our study was to explore the PCI-DCB-only approach as a strategy in two frequent situations: de novo and ISR lesions. These two conditions have completely different biological pathophysiological bases and natural histories. Therefore, it is inappropriate to compare the success of DCB in all contexts. Our goal, therefore, was to characterize when the PCI-DCB-only strategy is appropriate. Conceptually, we think that the adoption of DES as a fixed solution for all coronary stenoses, regardless of the vessel lesion etiology, might lead to TLF. This failure is attributed to the existence of tenacious metallic implants (stents) that interfere with vascular healing after PCI. Such interactions lead to chronic inflammation with subsequent formation of neo intimal atherosclerosis, a process that can facilitate TLF after stent-based PCI. Specifically, this subjacent milieu can explain our study findings. The more aggressive behavior in ISR cardiovascular death with respect to de novo lesions is possibly related to a greater atherosclerotic burden due the two atherosclerotic layers—the underlying native plaque and the superimposed formation of de novo atherosclerosis within the neointima, named “neoatherosclerosis”. We here report a combination of low TLR (6.7%) and bailout stenting (13.7%) rates after a DCB-PCI-only approach. The low rates of our findings are consistent with those found in other published registries [17,24,25,26,27], demonstrating a similar level of efficacy and safety. This low event rate is a result of the careful lesion preparation and DCB application that are consistent with recommended technique. The total MACE rates were high as a result of the high incidence of cardiovascular hospitalizations.

Other important findings in our study include the absence of lethal adverse complication vessel occlusion or stent thrombosis in the whole cohort of patients treated with the PCI-DCB approach. This finding has also been studied in other publications. Scheller et al. [28] observed no subacute thrombosis in their pivotal PACCOCATH trial after 24-month follow-up, but their study population included only ISR patients. On the other hand, Waksman et al. [26] studied the effect in de novo lesions in their Valentines II trial, with no vessel thrombosis in the mid-term follow-up, with a mean of 7.5 months. We have now reinforced this favorable clinical outcome with two types of lesions with respect to a mid-term follow-up. This safety point is more robust when considering that the PCI-DCB strategy implies the short-term DAPT recommendation to reduce patient exposure to aggressive anti-aggregate drugs and to promote less bleeding complications. In our opinion, this finding should be considered when patients with a high-risk bleeding profile need percutaneous interventions.

## 5. Study Limitations

The nature of retrospective analysis data has inherent weaknesses, including the absence of a matching or randomization methodology. Another weakness is the single-center patient inclusion. However, this last limitation also ensures minimal diversity of operator techniques and the control of strict adherence to recommendations. One other limitation is the lack of intracoronary images, which would allow us to better understand the success and failure of our procedures. However, these limitations are also counterbalanced by the single-center nature of the study, ensuring a homogeneous technical approach to the PCI.

## 6. Conclusions

This single-center, observational study presents data on the quotidian, prevalent, cath-lab intervention, highlighting the efficacy of a DCB-only PCI strategy in de novo and ISR coronary lesions. These results add to the accumulating evidence that a DCB-only strategy can serve as an alternative for a stentless treatment of suitable de novo coronary lesions. Moreover, the use of DCB can serve as a palliative bridge to CABG (coronary artery bypass graft) surgery, and the lack of stent implantation will help to facilitate the surgery. This efficient technology is underrepresented in daily practice despite having been present in the PCI arsenal for about a decade.

In summary, in this study, we have provided evidence that a mid-term DCB-PCI strategy taking into account the underlying pathology is associated with a low TLR. We therefore believe that our data support the concept of a DCB-only PCI strategy in suitable de novo and ISR coronary lesions should have more places in daily practice.

## Figures and Tables

**Figure 1 jcm-11-01859-f001:**
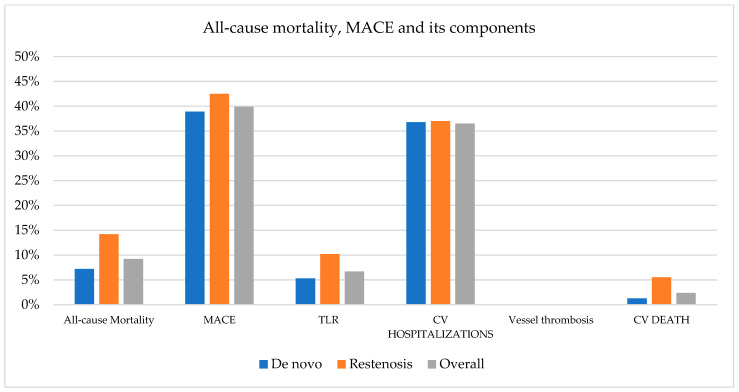
All-cause mortality, MACE and its components. MACE = major adverse cardiac events; TLR = target lesion revascularization; CV = cardiovascular.

**Figure 2 jcm-11-01859-f002:**
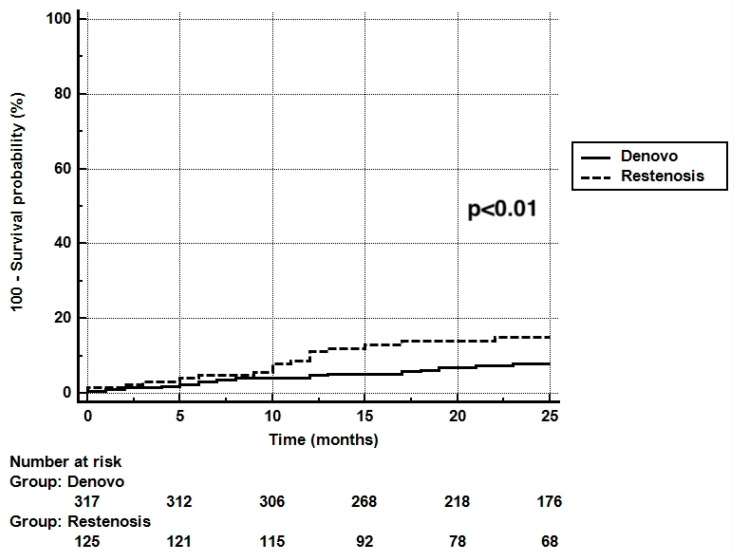
KM in-stent restenosis vs. de novo lesions.

**Figure 3 jcm-11-01859-f003:**
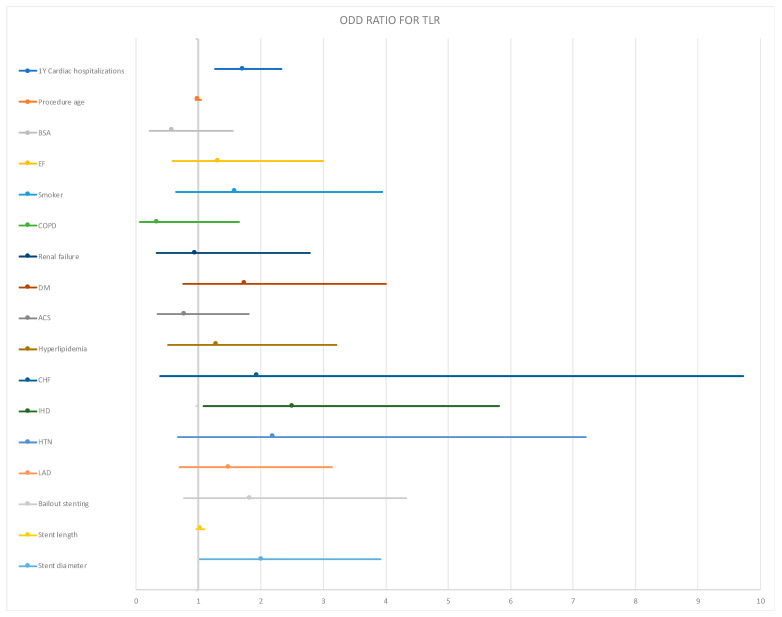
Subgroups analysis. BSA = body surface area; LVEF = left ventricle ejection fraction; COPD = chronic obstructive pulmonary disease; DM = diabetes mellitus; ACS = acute coronary syndrome; HF = heart failure; IHD = ischemic heart disease; HTN = hypertension; LAD = left anterior descending artery.

**Table 1 jcm-11-01859-t001:** Patient characteristics.

	De Novo	Restenosis	Total	*p*-Value
Patients *n* (%)	315 (71.3%)	127 (28.7%)	442 (100%)	-
Male *n* (%)	255 (81.0%)	94 (74.0%)	349 (79.0%)	0.136
Age	65.85 ± 11.59	66.21 ± 11.94	66.21 ± 11.69	0.317
BSA (m^2^)	1.89 ± 0.31	1.79 ± 0.38	1.86 ± 0.33	0.002
Diabetes mellitus *n* (%)	138 (43.3%)	64 (50.4%)	202 (45.3%)	0.172
Hypertension *n* (%)	224 (70.4%)	95 (74.8%)	319 (71.7%)	0.356
Hyperlipidemia *n* (%)	219 (68.7%)	103 (81.1%)	322 (72.2%)	0.008
Smoker *n* (%)	120 (37.6%)	49 (38.6%)	169 (37.9%)	0.850
Renal failure *n* (%)	41 (12.9%)	28 (22.0%)	69 (15.5%)	0.015
Known IHD *n* (%)	89 (27.9%)	53 (41.7%)	142 (31.8%)	0.005
Known HF *n* (%)	10 (3.1%)	10 (7.9%)	20 (4.5%)	0.028
COPD *n* (%)	23 (7.2%)	7 (5.5%)	30 (6.7%)	0.518
ACS *n* (%)	116 (36.4%)	40 (31.5%)	156 (35.0%)	0.331
LVEF < 50 *n* (%)	177 (55.5%)	86 (67.7%)	263 (59.0%)	0.018

BSA = body surface area; IHD = ischemic heart disease; HF = heart failure; COPD = chronic obstructive pulmonary disease; ACS = acute coronary syndrome; LVEF = left ventricle ejection fraction.

**Table 2 jcm-11-01859-t002:** Procedure data.

		De Novo *n* = 315	Restenosis *n* = 127	Total *n* = 442	*p*-Value
Reference vessel diameter (mm)		2.51 ± 0.53	2.97 ± 0.58	2.64 ± 0.59	<0.001
Vessel diameter <3.0 mm		227 (72.1%)	41 (32.3%)	268 (60.6%)	
Lesion length (mm)		22.77 ± 6.65	22.23 ± 6.16	22.62 ± 6.52	
Artery	LAD	135 (42.5%)	51 (41.8%)	186 (42.3%)	
	LCX	116 (36.5%)	34 (27.9%)	150 (34.1%)	
	RCA	50 (15.7%)	26 (21.3%)	76 (17.3%)	
	RI	16 (5.0%)	5 (4.1%)	21 (4.8%)	
	LMCA	1 (0.3%)	6 (4.9%)	7 (1.6%)	
Bailout stenting		55 (17.3%)	6 (4.7%)	61 (13.7%)	<0.001
Number of DCB used	1	299 (94.3%)	112 (90.3%)	411 (93.2%)	
	2	14 (4.4%)	11 (8.9%)	25 (5.7%)	
	3	4 (1.3%)	1 (0.8%)	5 (1.1%)	
Semi-complaint balloon		196 (62.2%)	47 (37.0%)	243 (55.0%	
Non -compliant balloon		102 (32.2%)	60 (47.3%)	162 (36.6%)	
Scoring balloon		6 (1.9%)	5 (3.9%)	11 (2.5%)	
Cutting balloon		11 (3.5%)	15 (11.8%)	(5.8%)	

LAD = Left anterior descending; LCX = Left circumflex; RCA = Right coronary artery; RI = Ramus intermedius; LMCA = left main coronary artery; DCB = Drug-coated balloon.

**Table 3 jcm-11-01859-t003:** Two-year clinical outcomes.

	De Novo	Restenosis	Overall	Log Rank *p*-Value
All-cause Mortality	7.2%	14.2%	9.2%	0.022
MACE	38.9%	42.5%	39.9%	0.478
TLR	5.3%	10.2%	6.7%	0.063
CV hospitalizations	36.8%	37.0%	36.5%	0.971
Vessel thrombosis	0	0	0	
CV death	1.3%	5.5%	2.4%	0.009

MACE = major adverse cardiac events; TLR = target lesion revascularization; TV-MI = Target vessel myocardial infarction; CV = cardiovascular.

## Data Availability

The data presented in this study is available on request from the corresponding author.

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
