# Peer review of "Mid-Term Clinical Outcomes Following Drug-Coated Balloons in Coronary Artery Disease"

_jcm, 2022, doi:10.3390/jcm11071859_

Round 1

Reviewer 1 Report

Dear authors,

Thanks for submitting your manuscript.

Use of intracoronary imaging would be nice to know particularly for ISR

In your introduction the messages are too wide

Please provide data on DEB (from which period is this available, indication in European guidelines etc …)

Please provide references for your method paragraph (classification of outcomes)

One of the main limitation is the very low use of imaging

Multivariable analysis are necessary here to know if DEB is associated with outcomes

Did you think about performing a matched comparison ?

Author Response

Dear reviewer,

Thank you very much for reviewing our manuscript. Your comments are much appreciated and were addressed in italics below point-by-point.

1) Use of intracoronary imaging would be nice to know particularly for ISR

As stated in the text, generally, intracoronary imaging is not routinely use in our institution, hence the low prevalence of its usage.

2) In your introduction the messages are too wide

We've edited the introduction accordingly and narrowed it down.

3) Please provide data on DEB (from which period is this available, indication in European guidelines etc …)

We've added the required data, now mentioned in last sentence of introduction.

4) Please provide references for your method paragraph (classification of outcomes)

References were added accordingly.

5) One of the main limitation is the very low use of imaging

Same as for question 1. We are aware of this limit and acting to increase the the usage of this tool in the upcoming years. 

6) Multivariable analysis are necessary here to know if DEB is associated with outcomes. 

Please refer to the figure attached which is also found in the manuscript. The multivariable analysis did not supply any additional significant information for our opinion and therefore we've decided not expand the discussion over it.

7) Did you think about performing a matched comparison ?

We didn't reckon that a matched comparison between different angioplasty techniques was needed as we didn't want to find out which method is better. Our aim was to investigate the long term outcomes of this particular technique. We will take this point to our attention in the future articles and sub analysis concerning DCBs.

Reviewer 2 Report

Interesting publication. Historically, PCI (formerly called PTCA) relied on coronary artery plasty with a calibrated balloon. This procedure was only effective in short-term observation. Only the introduction of stents prevented the rapid recurrence of stenosis. The authors discuss the disadvantages of using metal stents (coated or not).

The authors proposed a PCI technique using a coated balloon, obtaining similar results in de novo lesions and in lesions in previously implanted stents.

Currently, the DEB technique is used sporadically in some centers almost exclusively as a palliative bridge for cardiac surgery.

The presented work could result in a wider introduction of the DEB technique. This is supported by shortening the time of DAPT administration and facilitating CABG without the presence of metal stents. For this reason, the DEB technique will meet with the approval of the cardiac surgery environment.

The work requires minor adjustments. Inclusion and exclusion criteria should be clearly  listed. A two-year observation period cannot be called a long term. At most, mid term.

Author Response

Dear reviewer,

Thank you very much for reviewing our manuscript. Your comments are much appreciated and were addressed in italics below point-by-point.

1) This is supported by shortening the time of DAPT administration and facilitating CABG without the presence of metal stents. For this reason, the DEB technique will meet with the approval of the cardiac surgery environment.

1) This insight is now included in the "conclusion" section.

2) inclusion and exclusion criteria should be clearly listed.

2) We've re-arranged the manuscript titles and orientation and inclusion/exclusion criteria can now be found under section 2.1 "Patients population, inclusion and exclusion criteria".

3) A two-year observation period cannot be called a long term. At most, mid term.

3) Most of the studies done on DCB's included a follow up period of 6-12 month therefore we considered our follow up time as long term. We do agree with your terminology for the follow up period, therefore we've changed the title to "Mid term clinical outcomes following drug coated balloons in coronary artery disease".

Reviewer 3 Report

Sella et al present an interesting analysis on DEB treatment comparing ISR and de-novo lesions. My comments are as follows:

-Were the bail out -stented patients included in the outcome analysis and if. so, why? In my opinion, it would be sufficient to only report the percentage of bail out stented cases to indicate the rate of DEB strategy failure (which by in my opinion is closely related to operator experience with DEB).

-page 4, row 78. Captial A.

-page 7,  rows 143-145: 

"One third of the DEB angioplasty was done in the clinical setting of an ACS-non-ST elevation MI or unstable angina pectoris. In 116 patients (35.3%), DEB angioplasty was done in the clinical setting of an ACS "

Not understandable,  please clarify - which definition of ACS is used? NSTEMI+STEMI? 35.3% in the text and 36.4% in Table1? 

-figure 2 done in a hurry?: quality low, scaling unadjusted, "Restenisis" etc.

-Conclusions: I agree with the concept of potential benefit of stentless treatment of de novo lesions, but still the conclusion that this study has shown efficacy of DEB-treatment is unjustified as there was no non-DEB comparator. At least a comparison with event rates in modern DES-treated patients should be presented to back this up. Otherwise rephrase.

Author Response

Dear reviewer,

Thank you very much for reviewing our manuscript. Your comments are much appreciated and were addressed in italics below point-by-point.

1) Were the bail out -stented patients included in the outcome analysis and if. so, why? In my opinion, it would be sufficient to only report the percentage of bail out stented cases to indicate the rate of DEB strategy failure (which by in my opinion is closely related to operator experience with DEB).

1) We've included the bailout stenting in the outcomes as the DCB strategy was at first the intention to treat. Moreover, After primarily inflation of the DCB, the active drug paclitexal was deployed to the endothelium, therefore its future effect can be taken into account in our point of view, although bailout stenting was subsequently performed.   

2) page 4, row 78. Captial A.

2) Corrected.

3) page 7,  rows 143-145: 

"One third of the DEB angioplasty was done in the clinical setting of an ACS-non-ST elevation MI or unstable angina pectoris. In 116 patients (35.3%), DEB angioplasty was done in the clinical setting of an ACS "

Not understandable,  please clarify - which definition of ACS is used? NSTEMI+STEMI? 35.3% in the text and 36.4% in Table1? 

3) Typing mistake, corrected and cleared in the text. 

4) figure 2 done in a hurry?: quality low, scaling unadjusted, "Restenisis" etc.

4) We've corrected the figure accordingly. 

5) Conclusions: I agree with the concept of potential benefit of stentless treatment of de novo lesions, but still the conclusion that this study has shown efficacy of DEB-treatment is unjustified as there was no non-DEB comparator. At least a comparison with event rates in modern DES-treated patients should be presented to back this up. Otherwise rephrase.

5) The aim of our investigation was not to evaluate which revascularization technique is better, but was to evaluate the long (now revised to mid) term outcomes of this particular technique,  therefore we did not make any comparison. 

Round 2

Reviewer 1 Report

Still i feel the introduction is too general and do not focus enough on the problem

Author Response

Dear reviewer,

The introduction had been re-written to better focus on the main idea of this manuscript, according to  your guidance.  

Kindest regards,